# Neutron-Absorption Properties of B/Cu Composites

**DOI:** 10.3390/ma16041443

**Published:** 2023-02-08

**Authors:** Haoran Wang, Shuo Zhao, Junqing Han, Yuying Wu, Xiangfa Liu, Zuoshan Wei

**Affiliations:** 1Key Laboratory of Liquid-Solid Structure Evolution and Processing of Materials, Ministry of Education, Shandong University, Jinan 250061, China; 2Shandong Key Laboratory of Advanced Aluminum Materials and Technology, Binzhou Institute of Technology, Binzhou 256600, China

**Keywords:** copper matrix composites, Cu-B alloy, neutron absorption, boron nanosheets, high hardness

## Abstract

Copper has high electrical and thermal conductivity, which is frequently employed in structural and functional materials. In this research, powder metallurgy was used to incorporate boron nanosheets into metal matrix composites to create boron dispersion-enhanced copper matrix composites. The neutron-absorption characteristics of composite materials were investigated, as well as the link between neutron-absorption cross-section and neutron energy. The results told us that the morphology of the second phase on the particle surface is closely related to the size of Cu-B particles, copper and boron correspond atomically to each other on the interface without dislocation or lattice distortion, forming a completely coherent interface, and that the neutron absorption cross-section decreases exponentially as neutron energy increases. In low-energy neutrons with energies less than 0.1 eV, the increase of boron content and ^10^B abundance in Cu-B alloy will enhance the neutron-absorption capacity of the alloy. Boron dispersion-strengthened copper matrix composites have good neutron-absorption capacity, and the microstructure and size of boron do not affect the neutron-absorption performance of composites with the same content of boron. The hardness of the B-dispersion-strengthened Cu matrix composite obtained by nanoindentation test is about 3.04 GPa. Copper matrix composites with boron dispersion reinforcement exhibit high hardness and neutron-absorption characteristics.

## 1. Introduction

Radiation shielding is becoming increasingly important as the nuclear industry develops. Rays of all types, including gamma-ray, neutrons, and other charged particles, are exceedingly dangerous to the human body. Because neutrons are not influenced by electromagnetic force when they penetrate matter, they can only interact with atomic nuclei and so have a significant penetrating force. Lead is the most often used radiation-shielding material; however, it is very poisonous and has drawbacks such as high density, low flexibility, and limited shielding performance. Chen et al. [1] introduced the research progress of aluminum-based boron carbide (B_4_C/Al) neutron-absorbing materials and the advantages and disadvantages of different preparation methods. Chen et al. [2] simulated the neutron-shielding process of TPX/B_4_C composites by Monte Carlo method, and the results are compared with the boron polyethylene, lead boron polyethylene, and aluminum-based boron carbide. Zhao et al. [3] proposed a method for testing neutron-shielding performance for thermal, epithermal, fast neutrons, and neutron ambient dose equivalent rate by employing different response-tuning assembly (RTA). Cu is frequently employed in structural and functional materials. However, due to the weak mechanical characteristics of pure Cu, its future application is limited; thus, the augmentation of second phase particles has become an essential way of improving its mechanical qualities [4]. B has a high capacity for neutron absorption and can be employed as a reinforcing phase in composite materials to improve mechanical qualities. The B-reinforced Cu matrix composite is a suitable neutron-absorption material with a wide range of applications in nuclear power equipment, such as nuclear fuel workstations and pressure vessels. Jenei et al. [5] created carbon nanotube-enhanced Cu matrix composites. Wu TG et al. created Cu-based composites reinforced with a variety of carbon compounds and found outstanding mechanical characteristics [6]. B contains three valence electrons and a “three-center electron-missing bond” atomic structure [7]. ^10^B has a high neutron-absorption capability and has several applications in nuclear energy, national security, and medicinal therapy [8]. ^10^B may combine with neutrons to make ^11^B, based on the nuclear capture and fission reactions that occur when ^10^B is irradiated with thermal neutrons and fission quickly produces a ^7^Li ion, a high-energy alpha particle, and a low-energy gamma-ray (thermal neutrons flux = 7.5 × 10^9^ n/cm^2^ sec) [9]. People have also suggested B neutron-capture therapy to cure cancer by utilizing this property of ^10^B [10]. Cu is a suitable neutron absorber material matrix [11].

In this research, powder metallurgy was used to incorporate boron nanosheets into metal matrix composites to create the boron dispersion-enhanced copper matrix composites.

## 2. Experiment and Discussion

### 2.1. Experiment

By using powder metallurgy, the atomized Cu-B powder was crushed and sintered in solid phase. B nanosheets were incorporated into metal matrix composites, yielding dense B dispersion-enhanced Cu matrix composites. The procedure is as follows: a Cu-B alloy with a mass fraction of 2 wt.% B is atomized into Cu-B powder, ball milled, prepressed into shape in a hydraulic press, and then placed in a sintering furnace for solid phase sintering, resulting in the successful formation of a B diffusion-strengthened Cu matrix composite. The melt technique is used in the following manner: place the weighed high-purity Cu in a graphite crucible, melt it in a high-frequency induction furnace, raise the temperature to 1200 °C, add the high-purity Al in proportion, stir with a graphite rod until the melt is evenly mixed, and then add the quantitative B_2_O_3_ to the melt in batches for the reduction reaction. B-rich compounds can be formed by reducing the reactions that occur in the Cu-Al melt: 2Al+ B_2_O_3_ → 2B+ Al_2_O_3_, hold the heat briefly after the reaction is complete. Since the density of Al_2_O_3_ is lower than that of Cu, most Al_2_O_3_ floats on the surface of Cu melt, then remove the alumina on the surface of the melt. Finally, the melt was poured into a graphite mold and slowly cooled and solidified. The melt technique was used to create Cu-2B, Cu-4B, and Cu-2^10^B alloys. Cu-2B and Cu-4B is a copper-boron alloy containing 2 wt.% and 4 wt.% boron, respectively. There is no burn loss when the Cu-B alloy is proportionally smelted in the vacuum furnace, so we replace the real composition of the alloy with the composition in the configuration; that is, 4 in Cu-4B is 4 wt.% boron content. Cu-2^10^B was synthesized from pure Cu and H_3_BO_3_ with 95% abundance, and a 2% B mass fraction. A 95% abundance means that the percentage of ^10^B in all the boron atoms in boric acid is 95%. We purchased boric acid with a high abundance of ^10^B to prepare Cu-2^10^B directly.

The neutron absorption experiment was performed on the SANS instrument of China Spallation Neutron Source (CSNS) in Dongguan, Guangdong Province. The power of the neutron source was 80 kW. The polychromatic neutrons with the wavelength λ_n_ in the range of 4.8–7 Å were selected for data analysis. The samples with the dimension of 20 mm × 20 mm were ground to 2000 grit SiC paper surface finish. The detector can detect the wavelength and number of neutrons, and then calculate the neutron energy by de Broglie wavelength formula. The neutron absorption rate of samples is determined after obtaining the curve that relates neutron energy to number and comparing it to the data without samples. The wavelength λ = h/p is called de Broglie wavelength, and the relation λ = H/p and f = E/h is called de Broglie relation. These relations are the same as for photons, but for particles E = (1/2) mv^2^ = p^2^/(2 m), so E = h^2^k^2^/(2 m) and λ = h/√ (2 mE). Target en (b), 1 b = 10^−24^ cm^2^, is the unit of microscopic cross section, which measures the likelihood that neutrons may interact with atoms. The amount of neutron absorption is proportional to the neutron beam intensity, target thickness, and target core density when the target area is constant. The composite material should be formed into a block sample with a diameter of at least 20 mm, with parallel and flat top and lower sides. A transmission system and a receiving system comprise the neutron-absorption performance test equipment. The transmitting system is in charge of emitting neutron beams of varying energies, which are produced by high-energy protons bombarding heavy atomic nuclei (such as tungsten, uranium, and so on), and the receiving system is in charge of sensing the neutron beams passing through the sample, in order to calculate the transmissivity of a specific sample to neutrons of varying energies and then obtaining the neutron absorption rate.

The sample was observed by scanning electron microscope (SEM) of model SU-70. The maximum acceleration voltage of SU-70 was 20 kV and the magnification was 20–200, 000 times. The microstructure of Cu-B powder is shown in Figure 1a. Cu-B powder has a diameter of about 50 μm, and a small number of particles with a diameter of about 10 μm form on the larger powder. As shown in Figure 1b, Cu-B atomized powder was ball-milled to minimize the size of the second phase, and shows the electron diffraction spots of B. Mechanical peeling, high temperature and high-pressure treatment, extraction, and filtration were used to create B nanosheets from Cu-B atomized powder [12]. As shown in Figure 1c, B nanosheets were subsequently inserted as the reinforcing phase into the Cu-based composites, and the B dispersion enhanced Cu-based composites were effectively manufactured by using the powder metallurgy process. 

### 2.2. Results and Discussion

The relationship between the neutron energy and the microscopic cross-section of ^10^B is depicted in Figure 1d. This data is obtained through the ENDF/B6 database, and after taking logarithms, the relationship is linear. The tiny cross-section drops negatively exponentially as neutron energy increases. Unlike the microscopic cross-section, the dimension of the macroscopic cross-section is the reciprocal of the length, usually in cm^−1^ units. The macroscopic cross-section, which is the sum of the tiny cross sections of all the nuclei in a cubic centimeter, represents the probability of a nuclear reaction occurring for every 1 cm of neutron transit in matter. The macroscopic cross-section and neutron energy similarly have a negative exponential relationship.

Figure 2 shows a comparison of the three alloys’ neutron-absorption characteristics. The relationship between neutron transmittance number and neutron energy is seen in Figure 2a. The purple curve represents the neutron transmittance number without samples, and the red, blue, and green curves represent the neutron transmittance numbers of Cu-2B, Cu-4B, and Cu-2^10^B. As seen in Figure 2b, as neutron energy increases, the sample’s neutron-absorption rate decreases. Cu-4B, with a thickness of approximately 0.4 cm, has an absorption rate of more than 80% for low-energy neutrons with energies less than 0.1 eV, as indicated in the blue curve. However, for neutrons with an energy of more than 10 eV, its absorption rate decreases to less than 35%. The sample thickness is homogenized to produce the macroscopic cross-section of the sample because the quantity of neutron absorption is proportional to the target thickness when the target area remains constant. When the ordinate is 1, the intersection points of the three curves and the abscissa are depicted by dotted lines of different colors, showing that the sample with a thickness of 1 cm can absorb practically all neutrons below this energy, as illustrated in Figure 2c. 

The macroscopic cross-section and sample energy have an equal connection log_10_^y^ = alog_10_^x^ + b, where x and y indicate neutron energy and macroscopic cross-section, respectively, and a and b are constants. By taking two points in the straight-line part of the data, the relation between the cross-section and energy of the sample can be fitted, so as to predict the neutron-absorption capacity of the sample at a certain energy. The fitting data of the three curves are as follows: ①:Cu-2B: log_10_^y^ = −0.14477log_10_^x^ – 0.25091, y = 0.56116x^−0.14477^②:Cu-2^10^B: log_10_^y^ = −0.2082log_10_^x^ – 0.024, y = 0.94624x^−0.2082^③:Cu-4B: log_10_^y^ = −0.21294log_10_^x^ + 0.02823, y = 1.0672x^−0.21294^.

As shown in Figure 3, under the same and lower neutron energy (<10 keV), Cu-4B has the strongest neutron-absorption capacity, followed by Cu-2^10^B, and Cu-2B is the weakest. It shows that the neutron-absorption capacity of Cu-B alloy grows with increasing B content, but the macroscopic cross-section does not increase linearly. Although increasing the abundance of ^10^B can improve the material’s neutron-absorption capability, the connection between the amount of ^10^B is also not linear. The ability to predict the neutron absorption of a sample at a given energy is provided by a material’s neutron absorption cross-section, which is negatively exponential with neutron energy. The macroscopic cross-sectional curve of neutron absorption of high purity Cu(99.9 wt.%)—neutron energy relationship is shown in Figure 3. To obtain the connection, the linear component of the curve is fitted: ④:log_10_^y^ = −0.11808log_10_^x^ − 0.3499, y = 0.44679x^−0.11808^.

For a neutron beam with an energy of 1000 eV, the macroscopic cross-section of pure Cu (99.9 wt.%) is approximately 0.198 cm^−1^, which is close to the macroscopic cross-section of three Cu-B alloys. At this time, the influence of the content and abundance of B on the neutron absorption capacity of the sample is very small.

B/Cu composites’ neutron-absorption qualities were evaluated to see if they could absorb neutrons well while maintaining outstanding mechanical properties. The effects of different B morphologies on the composites’ neutron absorption characteristics were also looked into. [13]. The powder is pressed into a block sample in a hydraulic press with a pressure of approximately 50 MPa and pressure holding time of approximately 20 s. The shaped sample is placed in a vacuum sintering furnace at 1000 °C for 1 h without pressure sintering, and then slowly cooled in the furnace with three kinds of composite materials.

As shown in Figure 4a–c, three different types of B-dispersion-strengthened Cu-based composites were created by hot pressing sintering in order to produce denser composites and various second-phase morphologies. As shown in Figure 4a, the bigger size B phase in the composites after mother alloy for atomization of Cu-B powder is spread in bulk over the matrix. Figure 4b shows the graphene oxide-coated Cu-B atomized powder, and a tiny quantity of B-phase coarsening composite material is obtained [14]. Cu matrix composites with B dispersion distribution were obtained by using Cu-B alloy powder after ball milling in Figure 4c, and the size of B was less than that of composites generated without press-sintering [15]. The relation between the macroscopic cross section of B dispersion-strengthened Cu matrix composite and neutron energy is shown in the illustration, which is indicated by green, red, and blue curves, respectively [16]. Figure 4d shows the comparison of the macroscopic cross sections of the three composite materials. The three curves almost overlap. After amplification, the macroscopic cross-sections of the three materials are slightly different. The neutron absorption capacity of the composite material that corresponds to boron nanosheets is somewhat better, followed by that of the composite material with local coarsening, and finally by the composite material with massive coarsening, which has the lowest neutron-absorption capacity.

The linear fitting of the straight-line part of the three curves is taken to obtain the prediction formula of sample macroscopic cross-section neutron energy. Figure 4a–c is Samples ⑤–⑦ respectively as follows: ⑤:log_10_^y^ = −0.19809log_10_^x^−0.00143, y = 0.99671x^−0.19809^⑥:log_10_^y^ = −0.19953log_10_^x^ + 0.00673, y = 1.01562x^−0.19953^⑦:log_10_^y^ = −0.19844log_10_^x^ + 0.01005, y = 1.0234x^−0.19844^.

Figure 4e shows the fibrous B, and Figure 4f shows the electron diffraction spot. After calibration and comparison with the PDF card (PDF#12-0377), the B phase is determined. Figure 4g is an enlarged view of the interface, revealing the interface bonding between fiber B and Cu matrix. The upper part of the picture is Cu matrix. The spacing of crystal planes shown in white lines is approximately 0.21 nm, corresponding to the Cu (111) plane. The lower part is divided into B, and the black line shows that the crystal plane spacing is also 0.21 nm, corresponding to B (202). Figure 4h shows B, and the angle between the two sides of the shape formed by the phase is 90°. The interfacial interaction between the B and Cu matrix is seen in Figure 4i by a solid blue line. The crystal plane spacing shown by the solid white line is approximately 0.21 nm, which corresponds to the Cu (111) plane. The plane spacing shown by the solid black line is approximately 0.44 nm, which corresponds to plane B (200). As shown in the Figure 4, Cu and B correspond atomically to each other on the interface without dislocation or lattice distortion, forming a completely coherent interface. Both B and Cu create a coherent relationship in the B dispersion reinforced Cu matrix composites, and the interface is well bound. [17].

With the same content of B, the morphology of B has little effect on the neutron absorption capacity of the material. Therefore, the neutron-absorption properties of the same composites are related to the content of B, but not to the morphology and size of B. The B form may have an effect on the total density and also cause differences in neutron absorption. Compared with the high B steel, Cu matrix has better electrical and thermal conductivity, and B does not form continuous network in Cu, and the composites’ excellent mechanical characteristics are caused by the distributed dispersion of nano B. When the B content is the same, Cu matrix composites strengthened by B dispersion have a superior neutron-absorption capacity [18]. In summary, B dispersion strengthened Cu matrix composites have high neutron absorption, both B and Cu form a coherent relationship, and the interface is well bonded, giving it broad application prospects.

We take the middle part of the sample for polishing slice. In the sheet sample, the size of boron particles is small, and the pressing depth range of nanoindentation is large. Moreover, due to the size effect of metal materials, the hardness and modulus tend to be a constant value with the increase of the pressing depth of the indenter, which is considered as the hardness of the material. Nanoindentation can also characterize hardness to some extent, so we choose nanoindentation for hardness test. We do nanoindentation tests on the same sample, and we can only measure data from one point at a time to get a group of data. Figure 5a shows the hardness of various copper matrix composites. When the indenter first touches the sample, the hardness value of the material is relatively low. With the increase of the pressing depth, the hardness increases, and tends to be fixed when the pressing depth reaches 40 nm. Therefore, we take the average value of all the data between 50 and 100 nm. The nanoindentation method was used to test relevant data of 90 points on the sample, among which relevant data of 89 points were valid and relevant data of 1 point was missing. Then the hardness of the sample was obtained by calculating the average value of relevant data of 90 points. The relation curve between the contact stiffness of the test point, and the pressing depth is shown in Figure 5b. Contact stiffness refers to the ability of the material to resist deformation under the action of force. With the increase of pressing depth, the contact stiffness of the test point increases linearly, and the resistance to deformation becomes stronger. Figure 5c shows the depth-elastic modulus curve of the test point. When the indenting head just touches the sample, the elastic modulus of the material is relatively low. With the increase of the indenting depth, the elastic modulus increases obviously, and tends to be fixed when the indenting depth reaches 20 nm. The elastic modulus is 113.66 GPa. Jenei et al. prepared carbon nanotubes reinforced with Cu matrix composites and tested their hardness by nanoindentation and achieved 2.5 GPa [5]. The hardness of B-dispersion strengthened Cu matrix composite obtained by nanoindentation test is approximately 3.04 GPa, which is higher than that of high-density nanotwin Cu (~2.7 GPa) [19], and also higher than that of other Cu matrix composites [20,21]. In summary, the B dispersion reinforced Cu matrix composites prepared by powder metallurgy method have high hardness.

## 3. Conclusions

With an increase in neutron energy, the cross-section of neutron absorption falls off exponentially. In low-energy neutrons with energies less than 0.1 eV, the increase of B content and ^10^B abundance in Cu-B alloy will enhance the neutron absorption capacity of the alloy. B dispersion-strengthened Cu matrix composites have good neutron-absorption capacity, and the microstructure and size of B do not affect the neutron-absorption performance of composites with the same content of B. Cu matrix composites with B dispersion reinforcement exhibit neutron-absorption characteristics, both B and Cu form a coherent relationship, and the interface is well bonded. Although a nanoindentation test shows that the hardness of Cu matrix composites with B dispersion reached 3.04 GPa, it is a material with high hardness.

## Figures and Tables

**Figure 1 materials-16-01443-f001:**
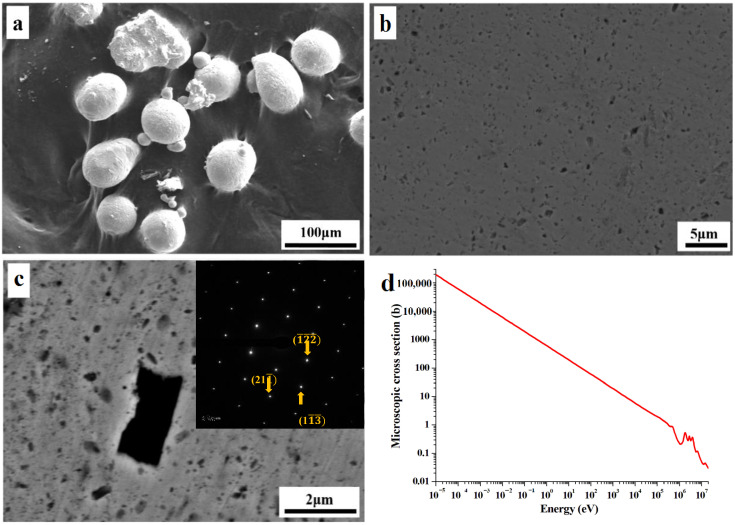
(**a**) Morphology and size of atomized Cu-B powders; (**b**) microstructures of cross-section of ball-milled powders and electron diffraction spot of B; (**c**) composite sintered by ball-milled powders; (**d**) neutron energy-microscopic cross-section curve of ^10^B.

**Figure 2 materials-16-01443-f002:**
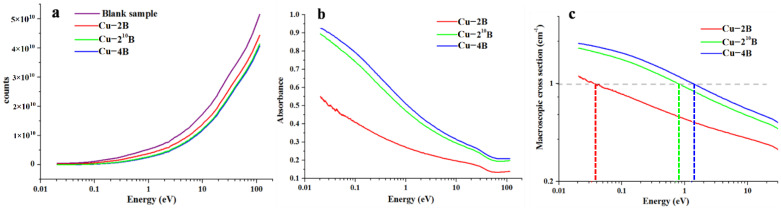
Neutron absorption properties of Cu-B alloy with different components (**a**) energy-neutron counts curves; (**b**) energy-neutron absorbance curves; (**c**) energy-macroscopic cross-section curves.

**Figure 3 materials-16-01443-f003:**
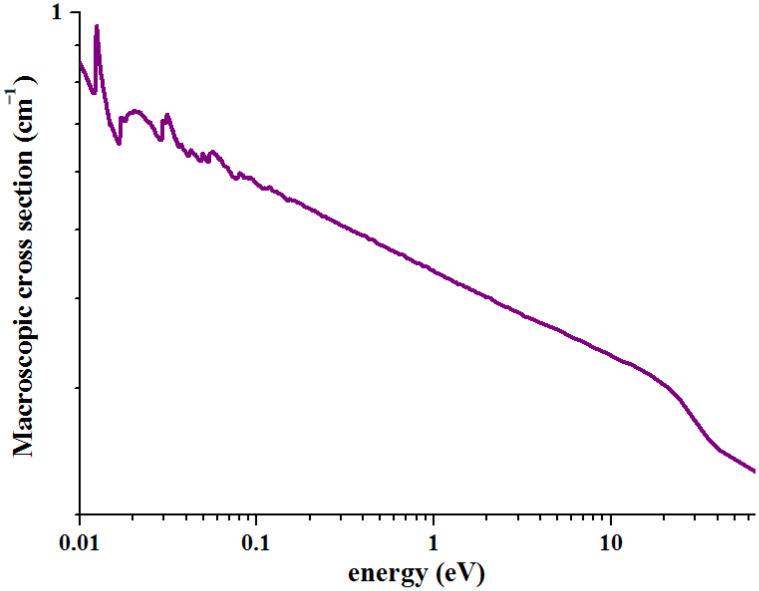
Energy-macroscopic cross-section curve of high purity Cu (99.9 wt.%).

**Figure 4 materials-16-01443-f004:**
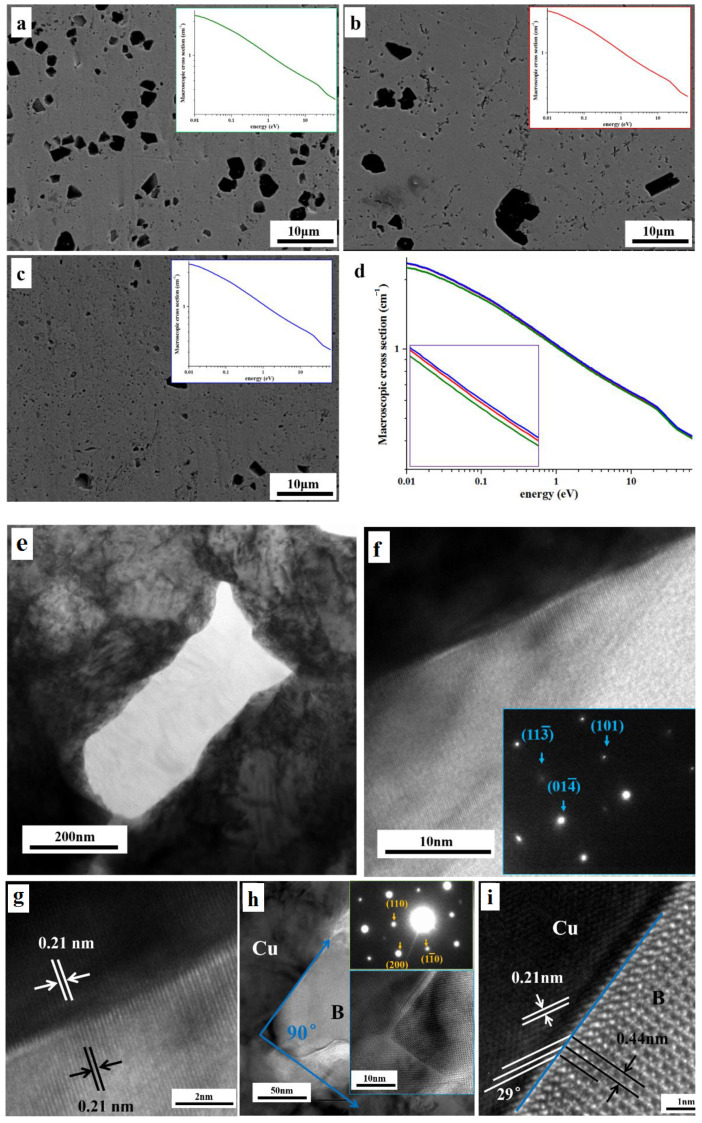
(**a**−**c**) The microstructure and macrosection of the composites prepared by atomization, graphene coating, and ball milling of Cu-B powder, respectively. (**d**) The comparison of macroscopic cross section of several composites shows that the neutron-absorption capacity is independent of the size and morphology of B. (**e**) Fibrous B. (**f**) Electron diffraction spot of B. (**g**) There exists a coherent interface relationship between B and Cu. (**h**) Boron and its electron diffraction patterns. (**i**) There is a coherent interface between boron and Cu.

**Figure 5 materials-16-01443-f005:**
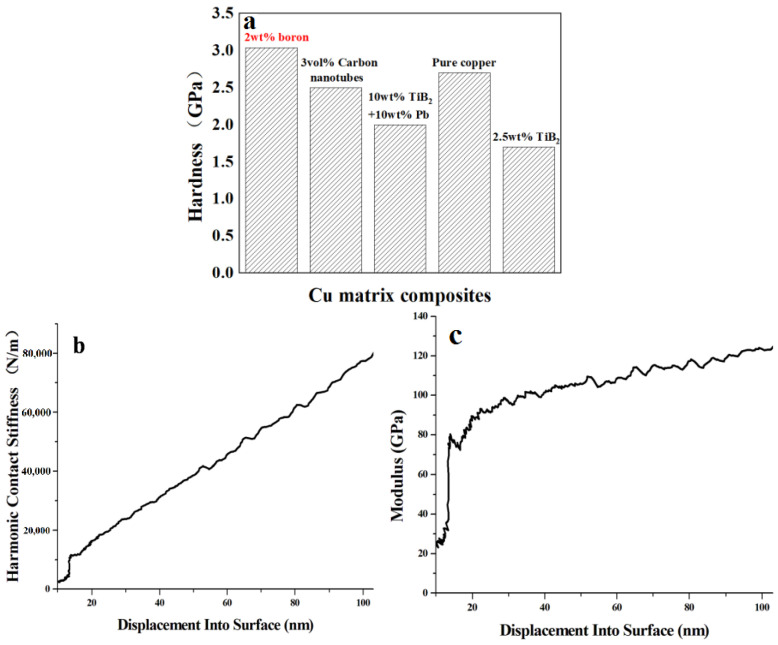
(**a**)The hardness of other Cu matrix composites reported previously and this work [5,19,20,21]; (**b**) displacement into surface-harmonic contact stiffness curve; (**c**) displacement into surface-elastic modulus curve.

## Data Availability

The data presented in this study are available in article.

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
