# Peer review of "Neutron-Absorption Properties of B/Cu Composites"

_materials, 2023, doi:10.3390/ma16041443_

Round 1
Reviewer 1 Report
This paper was already submitted to materials and was not changed after previous rejection (except for a few words in the results and conclusions sections).
The part about neutron experiments is poorly described. In my opinion the description of the measurement representing the core part of the paper can not be overlooked. This includes both the experimental setup and the description of quantities measured that must be given completely via equations that avoid any possible confusion stemming from the language.
Some of the results mentioned in the paper are not supported by the data, such as the enhancement of mechanical properties.
The whole paper is based on the measurement of neutron
attenuation but the only description of the experiment is "The detector can
detect the number and wavelength of neutrons". Clearly this is not
acceptable. The description of the neutron source must be given, including
where it is located and how it is operated. The geometry of the experiment
must be given (size of the neutron "beam" with respect to the composite
target, distance from the detector). The details of the detector must be
given: what kind of technology it uses, how large it is, etc.
Fast neutrons are usually identified as those with energy higher
than 100keV, about 4 orders of magnitude larger than the range explored in
this paper.
For the above reason my suggestion is to reject the paper.
Reviewer 2 Report
The work entitled: “Neutron absorption properties of B/Cu composites” is dealing with the formation of a Boron enhanced copper matrix and its neutron absorption properties.
The essence of investigation, the composition of the materials and the applicability are in the scope of the journal.
I recommend the following changes in the manuscript:
- Abstract, lines 10-14– the authors used information from other articles, but there are no citations of these studies. Please, add citations or (preferentially) remove this part of the abstract.
- Abstract, line 23 – 10B should be 10B (superscript).
- Please unify the symbols – Greek letter and spelled words for gamma rays (lines 32 and 59)
- Line 59 – please change “Li7+”
- Lines 77 and 81 – please use the correct chemical formula for boron anhydride and boric acid
- Line 78- please add information, on how the progress of the reaction was monitored as “after the reaction was complete” is not clear
- Line 149 – If we suppose the minimal value of refraction (as mentioned on line 110), both graphs for Transmittance (fig. 2b) and Absorbance (Fig. 2c) are probably unnecessary
- Lines 158-160 and 197-199 – please comment on the values of fitted expressions.
Reviewer 3 Report
The topic of the paper is about a new material designed to face neutron irradiation. The authors have elaborated different copper samples containing different content of natural boron or B-10. Their aim is to evaluate the neutron absorption properties of these samples.
My major concern is about the details of the experiments (and MCNP calculations). For instance,they should explain clearly the meaning of Cu-4B and Cu-2(10)B. Is "4" standing for the 4% of boron ? Is "95% abundance" standing for the boron-10 isotope ? If so, how was done the enrichment ? (chemical exchange distillation, cryogenic distillation or ion exchange resin method ?). The same goes for the MCNP calculations. More details are required. Which type of detector is used ? What about the microscope which was used for the images ? What about the details for the HP sintering (temperature ; pressure ; duration) ? and so on...
Also, the reader knows nothing about the mechanical properties which could have been deteriorated by the elaboration process of the materials. Some words should be added about it in order to know if the materials could be suitable for something.
That's why I recommand major corrections.
A minor concern is about the use of the symbols. Please avoid beginning a sentence with a symbol in particular in the abstract (for instance : Copper and not Cu, Boron and not B, …). Also, Li7+ has to be modified line 59).
Round 2
Reviewer 1 Report
After the addition of new information and data the paper is certainly improved.
However:
-the language should be strongly revised to ensure adequate understanding by the readers
-it is not clear what the mcnp simulation are needed for, and which of the shown results are obtained by simulation (possibly fig 1d?). This must be made clear.
-the addition regarding the instrumentation (lines 111-130 circa) that are now in section 2.2 should be in the experimental part (section 2.1)
-in the mechanical characterization part: what is a "group of data?" is the measurement performed on only one sample in different positions or in several samples? Since you collected a nice statistics you should give in addition to the average value also its standard deviation or some other information regarding the width of the distribution. This allows understanding if the difference between compositions is significant or not.
-the last sentence of the conclusion should be rewritten, as it is now it is difficult to understand what the authors mean.
Reviewer 3 Report
Thanks for the answers to my questions with full details as required.
Author Response
Thanks a lot for having reviewed our manuscript. We appreciate the reviewers’ kind work and their positive comments on the manuscript genuinely.